# Parent-mediated interventions versus usual care in children with autism spectrum disorders: A protocol for a systematic review with meta-analysis and Trial Sequential Analysis

**Charlotte Engberg Conrad**[1]*, **Sonja Martha Teresa Ziegler**[2,3], **Niels Bilenberg**[2,3], **Jens Christiansen**[4], **Birgitte Fagerlund**[5], **Rikke Hermann Jakobsen**[6], **Pia Jeppesen**[7], **Caroline Barkholt Kamp**[6,8], **Per Hove Thomsen**[9,10], **Janus Christian Jakobsen**[6,8], **Marlene Briciet Lauritsen**[1,11]

**1** Psychiatry, Aalborg University Hospital, Moelleparkvej, Aalborg, Denmark, **2** Child and Adolescent Psychiatry, Mental Health Services in the Region of Southern Denmark, Odense, Denmark, **3** Department of Clinical Research, University of Southern Denmark, Odense, Denmark, **4** Master International A/S, Copenhagen, Denmark, **5** Mental Health Services in the Capital Region of Denmark, Denmark, **6** Copenhagen Trial Unit, Centre for Clinical Intervention Research, Capital Region of Denmark, Denmark, **7** Department of Child and Adolescent Psychiatry, Copenhagen University Hospital – Psychiatry Region Zealand, Roskilde, Denmark, **8** Department of Regional Health Research, The Faculty of Health Sciences, University of Southern Denmark, Denmark, **9** Department of Child and Adolescent Psychiatry, Aarhus University Hospital, Psychiatry, Denmark, **10** Department of Clinical Medicine, Aarhus University, Denmark, **11** Department of Clinical Medicine, Aalborg University, Denmark

* charlotte.conrad@rn.dk

## Abstract

### Introduction

Autism spectrum disorder encompasses diverse patterns of social communication and repetitive, restricted behaviours. Various interventions have been developed to reduce the negative consequences of this disorder and improve levels of functioning, and recently interest in parent-mediated interventions has increased. Previous reviews and meta-analyses have investigated the effects of the parent-mediated interventions, however; a systematic review with meta-analysis of high quality has not been performed since 2013. This protocol for a systematic review with meta-analysis aims to describe the methods and purpose of synthesising current evidence regarding the effects (both positive and adverse) of parent-mediated interventions in both children with autism and their parents.

### Methods

Electronic searches will be conducted in Cochrane Central Register of Controlled Trials (CENTRAL), Medical Literature Analysis and Retrieval System Online (MEDLINE), Excerpta Medica database (EMBASE), Latin American and Caribbean Health Sciences Literature (LILACS), American Psychological Association PsycInfo

**Data availability statement:** No datasets were generated or analysed during the current study. All relevant data will be made available upon study completion.

**Funding:** This work was supported by The Lundbeck Foundation, grant number R389-2021-1597, TrygFonden, grant number 155050, Novo Nordisk Foundation, grant number NNF22SA0081180, Karen Elise Jensen Foundation, and the Simon Fougner Hartmann Family Fund. The funders had no role in study design, data collection and analysis, decision to publish, or preparation of the manuscript.

**Competing interests:** The authors have declared that no competing interests exist.

(PsycInfo) and Science Citation Index Expanded (SCI-EXPANDED). Randomised clinical trials of parent-mediated interventions for children with autism and control groups of usual care, waiting list or no treatment will be included. Two reviewers will independently screen, select and collect data. Methodological quality of included studies will be evaluated using Cochrane methodology. The primary outcome will be autism symptom severity as measured by the Autism Diagnostic Observation Schedule. Secondary outcomes will be adaptive functioning, adverse effects, child language, child´s quality of life, parental quality of life and parental stress. Meta-analyses and Trial Sequential Analysis will be performed.

## Discussion

This is the study protocol for a systematic review and meta-analysis of parent-mediated interventions versus usual care for children with ASD. Results of the review will inform clinicians and parents about the current evidence of the effects, both positive and negative, of parent-mediated interventions on younger children with autism and their parents, through improved methodology and inclusion of new studies. PROSPERO registration number: 385188

---

## Introduction

### Description of the condition

Autism spectrum disorder (ASD) is a neurodevelopmental disorder affecting 1–2% of the population in high-income countries [1,2]. This disorder is characterised by impairment of social communication and repetitive restricted behaviour [3,4]. ASD is heterogeneous and encompasses diverse patterns of these symptoms [5]. Impairments in individuals who have ASD are pervasive and present across all areas of functioning; however, the association between ASD symptom severity and adaptive functioning remains unclear [6–8].

ASD is associated with a higher risk of comorbid medical and psychiatric disorders, e.g., epilepsy, tuberous sclerosis, depression, anxiety, attention deficit disorder and intellectual disability [4,9–13]. Consequently, individuals with ASD often experience reduced quality of life and lower educational levels, as well as a higher mortality rate [4,9]. Due to the significant functional impairments that can be experienced, ASD is associated with long-term health, social and financial costs for individuals, their families and society as a whole [14]. In the United States, the additional cost to teach an autistic child is $8,600 yearly [15,16]. Lifetime societal costs of ASD range between $1.4 and $2.4 million per individual. These costs include interventions, therapies, special education, productivity loss and adult care [17], comparable to costs found in other Western countries [18].

### Description of the intervention

To reduce the negative impact of ASD, various interventions have been developed to improve adaptive functioning and child and parental quality of life, as well as reducing

core autism symptoms [19–21]. Several studies have suggested that early intensive education and therapies might improve developmental outcomes in children with ASD [4,22,23].

**Behavioural interventions.** Behavioural interventions rooted in operant learning theory were some of the first interventions for children with ASD [23]. This theory is reflected in the field of applied behaviour analysis (ABA) and the systematic application of behavioural principles aimed at both improving socially significant behaviour and identifying variables that elicit behavioural change [24,25].

**Developmental approaches.** Developmental interventions are grounded in the belief that research-based knowledge of both normal and abnormal development may inform and enhance one another [26]. Studies have demonstrated that developmental trajectories in young children with ASD are similar to those of typically developing children within various developmental domains. This includes the development of the attachment system [27–31] and thus emphasises incorporating developmental principles and sequences in early autism treatment. Interventions within this framework focus on improving developmental aspects such as joint attention, synchrony, reciprocity, imitation, language and duration of parent-child or child-child interactions [23,28,32].

**Naturalistic developmental behavioural interventions.** Since the early 2000s, interventions rooted in both behavioural and developmental theory (i.e., Naturalistic Developmental Behavioural Interventions (NDBIs)) have emerged [33]. These interventions are based on both social learning theory and research in child development and demonstrate that these two fields can be integrated, and interventions incorporated that support the strengths offered by each perspective.

**Other interventions.** Other interventions for children with ASD do not fall into either behavioural or developmental frameworks and include—but are not limited to—sensory-based interventions, such as sensory integration therapy [34] and interventions that provide structure such as TEACCH (Treatment and Education of Autistic and related Communication-Handicapped Children) [35].

**The broad category of parent-mediated interventions for ASD.** The above four categories of interventions vary in intensity, form and by whom they are applied. For this review we will include behavioural, developmental, NDBI and other interventions when applied by therapists targeting primarily the parents, in groups or individually, in order to make parents the agents of change of the child's behaviour. In short, in this review and meta-analysis, the focus will be on published research where parents actively delivered interventions to the child.

In the early years of ASD interventions, parents were taught to use strategies based on learning principles to improve their children's behaviour at home [36]. In recent decades, however, an increased interest in targeting early parent-child interactions has been observed and parent-mediated interventions (PMI) – where the parent is the agent of change in the child's behaviour [37] – have been developed within different intervention frameworks.

PMI' benefits include reducing therapist time and costs, and treatment can be brought into the home and everyday life, thus promoting the generalisation of skills. The cost-effectiveness of PMI enables the adaption of interventions to various socio-economic and cultural contexts. Parents often seek tools to support the optimal development of their child with ASD effectively, and PMI may strengthen their sense of empowerment and self-efficacy, which may reduce parental stress [4,38,39]. As the primary participant in the intervention, parents can apply the intervention in a sensitive and individualised manner, which may also reduce demands on their child [40]. Individuals with ASD often have difficulty generalising skills learned in one setting to other settings or maintaining behavioural change [41,42]. PMI enable parents to continuously implement support strategies and teach skills in a variety of situations and settings throughout the day. Due to these advantages for the children, parents and society, this systematic review will focus on PMI.

## Why is it important to do this review

**Previous reviews of PMI.** Previous reviews and meta-analyses of PMI conducted with children with ASD have found evidence for a treatment effect on disruptive behaviour in children with ASD through parent training [43–46]. In a review

and meta-analysis examining parent-focused outcome measures, Rutherford et al. [47] indicated that PMI may reduce parent stress as well as improve parenting style and satisfaction. Conversely, a Cochrane review and meta-analysis [40] did not find a significant benefit of PMI in relation to parental stress and most other outcomes such as language, communication, and adaptive functioning, though positive child-focused outcomes of moderate size were found for child language comprehension and severity of autism characteristics. Several reviews examining child-focused outcomes support a positive change in child symptoms related to autism symptom severity, including improved child joint attention, language-communication and cognition [40,48,49]. Importantly, a recent review of PMI focusing on school-aged children with ASD suggests that the importance of using parents as mediators of change in interventions remains and possibly increases as children grow older [43]. In one review, Liu et al. [50] exclusively selected studies from China, Hong Kong and Taiwan to address this particular cultural setting. Liu et. al.'s review shows favourable effects of PMI, and the authors suggest an increased potential for implementing PMI in low-resource contexts. An overview of the previous reviews is presented in Table 1.

Although the evidence of parent-mediated, developmentally based interventions varies with stronger evidence for interventions such as Paediatric Autism Communication Therapy and less evidence for interventions such as Hanen´s More Than Words; the European Society of Child and Adolescent Psychiatry (ESCAP) recommends the use of PMI' main principles [51]. These principles include parental involvement, using naturalistic opportunities for learning during daily routines and facilitating the generalisation of skills across environments for improving social communication, especially in younger children [51,52]. Clinical guidelines from the National Institute of Health and Care Excellence (NICE) recommend considering a parent, caregiver or teacher mediation for psychosocial interventions [53]. The American Academy of Pediatrics states that including parents in interventions for children with ASD is critically important [54].

## The methodological quality of previous reviews

Previously performed reviews have concluded that the overall methodological quality of the included studies varies in terms of differences in study duration, sample size, and control condition employed. The reviews also vary in quality and methodology, which can be quantified using the AMSTAR 2 tool [55]. The first and second authors (CEC, SMTZ) of the present protocol scored the previous reviews independently using the [45,48] 2 tool (Table 2) and subsequently obtaining a consensus agreement. A third author (MBL) was consulted in case of disagreement, and agreement was reached through discussion. The reviews all had performed study selection in duplicate (AMSTAR item 5, Table 2) and used appropriate measures for statistical combination of results (item 11, Table 2). However, several of the other AMSTAR items were not fulfilled by some of the reviews, e.g., registration of a prespecified protocol (item 2, Table 2), providing a list of excluded studies and justify exclusion (item 7, Table 2), reporting on sources for funding in included studies (item 10, Table 2), providing explanation for any heterogeneity observed in the results (item 14, Table 2). Only one of the reviews was rated as high quality [40]. One review was rated low in quality [46] and the quality of the remaining seven reviews was rated as critically low [43–45,48–50,56]. In conclusion, a high-quality review of parent-mediated interventions has not been performed since 2013. Several studies have been published since, and it is time to conduct a new high quality systematic review with meta-analysis.

We previously performed the largest systematic review of PMI published to date. This review included a total of 1934 participants [46]. That particular review is the only systematic review of PMI that assesses adverse effects, though none of the included trials assessed adverse effects. The review showed mild to moderate support for PMI, and emphasised a need for studies of a higher quality with larger samples and lower risk of bias [46]. The review search was performed in March 2019, and it is thus expected that more trials have been published since. The need to implement a wider range of outcomes than the ones investigated in the earlier studies, such as joint attention, became evident while performing the previous review and will also be investigated in this search. In the present review, we will use Trial Sequential Analysis which has not been used in any of the previous reviews of PMI. Trial Sequential Analysis represents an analysis of

**Table 1.** Reviews of parent-mediated interventions for children with autism.

| First Author | Title | Design | No of trials | No of patients | Published protocol | Using GRADE[1] | Assessment of adverse effects | Assessment of risk of bias |
|---|---|---|---|---|---|---|---|---|
| Oono et al. (2013) | Parent-mediated early intervention for young children with autism spectrum disorders (ASD) | Cochrane review | 17 | 919 | No | Yes | No | Yes |
| Postorino et al. (2017) | A Systematic Review and Meta-analysis of Parent Training for Disruptive Behavior in Children with Autism Spectrum Disorder | Systematic review and meta-analysis | 8 | 653 | No | No | No | Yes – Using Cochrane risk of bias tool |
| Nevill et al. (2018) | Meta-analysis of parent-mediated interventions for young children with autism spectrum disorder | Systematic review and meta-analysis | 19 | 1205 | No | Yes | No | Yes |
| Althoff et al. (2019) | Parent-Mediated Interventions for Children with Autism Spectrum Disorder: A Systematic Review. | Systematic review | 13 | 835 | No | No; Used American Occupational Therapy Association (AOTA) guidelines | No | Yes – Using Cochrane risk of bias tool |
| Tarver et al. (2019) | Child and parent outcomes following parent interventions for child emotional and behavioral problems in autism spectrum disorders: A systematic review and meta-analysis | Systematic review and individual level meta-analysis | 11 articles, 9 trials | 521 | Registered on the Prospero database (registration number CRD42016033979) | No | No | Yes – Using Cochrane risk of bias tool |
| Rathliff-Black et al. (2021) | Parent-Mediated Interventions for School-Age Children With ASD: A Meta-Analysis. (NB Mean age 6–18) | Individual and group-level meta-analysis | 15 articles, 18 studies | 170 | No | No | No | No |
| Deb et al. (2020) | | Systematic review and meta-analysis | 17 articles, 15 trials | 975 | Research Registry Unique Identifying Number: review registry 915 | No | No | Yes – Using Cochrane risk of bias tool |
| Liu et al. (2020) | A systematic review and meta-analysis of parent-mediated intervention for children and adolescents with autism spectrum disorder in mainland China, Hong Kong, and Taiwan | Systematic review and meta-analysis | 12 trials | 964 | PROSPERO database (registration number CRD42019–138723). | Yes | No | Yes – Using Cochrane risk of bias tool |
| Conrad et al. (2021) | Parent-mediated interventions for children and adolescents with autism spectrum disorders: a systematic review and meta-analysis | Systematic review and meta-analysis | 30 trials | 1,934 | The protocol is registered at the Danish Health Authority website (www.sst.dk) | Yes | Yes | Yes |

[1]GRADE: Grading of Recommendations Assessment, Development and Evaluation system.

meta-analytic data with more transparent assumptions and better control of both false negative and false positive conclusions than traditional meta-analysis [57]. The objective of this protocol for a systematic review with meta-analysis is to describe the methods and purpose of synthesising current evidence of the effects – both positive and adverse – of PMI versus usual care on children with ASD and their parents, through improved methodology and inclusion of newly published studies.

Table 2. AMSTAR 2 Assessment of systematic reviews.

| Study | Amstar 2 checklist items | | | | | | | | | | | | | | | | Overall quality |
|---|---|---|---|---|---|---|---|---|---|---|---|---|---|---|---|---|---|
| | 1 | 2 | 3 | 4 | 5 | 6 | 7 | 8 | 9 | 10 | 11 | 12 | 13 | 14 | 15 | 16 | |
| Oono et al. (2013) | Yes | Yes | Yes | Yes | Yes | Yes | Yes | Partial Yes | Yes | Yes | Yes | Yes | Yes | Yes | Yes | Yes | High |
| Postorino et al. (2017) | No | No | No | Partial Yes | Yes | Yes | No | Yes | Yes | No | Yes | No | Yes | Yes | Yes | No | Critically Low |
| Nevill et al. (2018) | Yes | No | No | Partial Yes | Yes | Yes | No | Partial Yes | Partial Yes | No | Yes | No | Yes | Yes | No | Yes | Critically Low |
| Althoff et al. (2019) | No | No | No | Partial Yes | Yes | No | No | No | Yes | No | – | – | No | No | – | No | Critically Low |
| Tarver et al. (2019) | Yes | Yes | No | Partial Yes | Yes | Yes | No | Yes | Yes | No | Yes | No | No | Yes | No | No | Critically Low |
| Deb et al. (2020) | Yes | Yes | No | No | Yes | Yes | No | Yes | Yes | No | Yes | Yes | Yes | Yes | No | Yes | Critically Low |
| Liu et al. (2020) | Yes | Yes | Yes | Partial Yes | Yes | Yes | No | Yes | Yes | No | Yes | Yes | Yes | No | No | Yes | Critically Low |
| Rathliff-Black et al. (2021) | Yes | No | No | Partial Yes | Yes | Yes | No | Partial Yes | No | No | No | No | No | No | No | No | Critically Low |
| Conrad et al. (2021) | Yes | Partial yes | Yes | Partial Yes | Yes | Yes | Yes | Partial Yes | Yes | Yes | Yes | Yes | Yes | No | No | Yes | Low |

## Methods

The present protocol has been registered in the PROSPERO database (registration number 385188) and is being reported in accordance with the reporting guidance provided in the Preferred Reporting Items for Systematic Reviews and Meta-Analysis Protocols (PRISMA-P) statement [58,59] (see checklist in Supporting information S1 File).

### Inclusion and exclusion criteria for considering studies for this review

**Types of studies.** This systematic review with meta-analysis will include randomised clinical trials irrespective of trial design, setting, publication status and publication year; however, only trials in English, German, Spanish and Nordic languages will be considered due to the authors' language skills. Quasi-randomised trials or observational studies will be excluded. The reason for only including randomised controlled trials will be to ensure the highest level of evidence.

**Types of participants.** We will include children (as defined by trialists) of all ages with a confirmed clinical or research diagnosis of ASD according to DSM or ICD criteria and further participants meeting diagnostic criteria based on diagnostic measures in accordance with these. Additionally, we will include the youngest children (as defined by trialist) with symptomatology strongly suggesting ASD. Participants will be included irrespective of any comorbidities.

**Types of interventions.** Any type of PMI for children with ASD will be included. A PMI is defined as a skill-focused intervention where the parent is an agent of change, and the child is the direct beneficiary [37]. The intervention will be provided by the parent to the child supported by a therapist. The majority of the intervention time with the therapist should be directly with the parent to be included as PMI. PMI targeting core autism symptoms or child maladaptive behaviour will be included.

**Control group.** We will accept control groups of usual care, waiting list or no treatment but not a control group of active interventions. In usual care control groups parents are usually not restricted from having their child participate in other not parent-mediated interventions of their choice, e.g., speech therapy or behavioural therapy.

### Outcome measures

### Primary outcome measure

1. *Autism symptom severity* as measured by the clinician-rated Autism Diagnostic Observation Schedule (ADOS) [60] using the ADOS total score with continuous outcome scores.

**Secondary outcome measures**

1. *Adverse effects* as measured by the Negative Effects Questionnaire [61] as a valid continuous measure.

2. *Child adaptive functioning* as measured by any version of the parent-rated Vineland Adaptive Behavior Scale [62]. If a trial only reports scores of Vineland subdomain scores, we will contact the author to ask for access to the missing data.

3. *Child's quality of life* as measured by the parent-rated Pediatric Quality of Life Inventory [63] as a valid continuous measure.

4. *Child language* as measured by the Mullen Scales of Early Learning [64], a valid continuous measure.

5. *Parental quality of life* as measured by the World Health Organization Quality of Life Assessment-BREF [65] as a valid continuous measure.

6. *Parental Stress* as measured by the Parenting Stress Index any form [66] as a valid continuous measure.

**Exploratory outcomes**

1. *Disruptive behaviour* as measured by the following parent-rated continuous rating scales: Aberrant Behavior Checklist (ABC) [67], Home Situations Questionnaire [68], Child Behavior Checklist – EXT (CBCL) [69], Eyberg Child Behavior Inventory (ECBI) (intensity subscale) [70] or any other valid continuous measure.

2. *Joint attention* as measured by the clinician-rated structured observation Dyadic Communication Measure for Autism [71,72] or any other valid continuous measure.

3. *Child adaptive functioning* as measured by any parent-rated version of the Adaptive Behavior Assessment System [73,74] or any other valid continuous measure.

4. *Parent synchrony/sensitivity* as measured by the questionnaire Emotional Availability Scale [75] or any other valid continuous measure.

5. *Attachment* as measured by Maternal Perception of Child Attachment [76], The Experiences in Close Relationship Scale [77] or any other valid continuous measure.

6. *Parent fidelity* to intervention implementation as measured explicitly by macro- or micro-codes of any or all of the following: dosage, adherence, quality, responsiveness, differentiation [78,79] or any other valid continuous measure

7. *Family life functioning* as measured by the Family Assessment Device [80,81], Autism Family Experience Questionnaire (AFEQ) [82] or any other valid continuous measure.

8. *Autism symptom severity* as measured by: ADOS calibrated severity score (CSS) [60,83], the clinician-rated scales Childhood Autism Rating Scale (CARS) [84], Autism Behavior Checklist [85], the systematic parental interview Autism Diagnostic Interview [86], the parent-rated Social Responsiveness Scale [87], Pervasive Developmental Behavior Inventory [88], Autism Behavior Checklist [85], or any other valid continuous measure of autism symptom severity.

9. *Child social communication symptoms* as measured by the Clinical Global Impression Scale [89], ADOS Social Affect subdomain [60] or any other valid continuous measure.

10. *Child repetitive behaviour* as measured by the ADOS Restricted and Repetitive Behavior subdomain [60] or any other valid continuous measure.

**Assessment time points.** The primary assessment time point will be the time-point closest to the end of treatment as defined by trialists.

## Search methods for identification of studies

**Electronic searches.** We will search Cochrane Central Register of Controlled Trials (CENTRAL), Medical Literature Analysis and Retrieval System Online (MEDLINE), Excerpta Medica database (EMBASE), Latin American and Caribbean Health Sciences Literature (LILACS), American Psychological Association PsycInfo (PsycInfo) and Science Citation Index Expanded (SCI-EXPANDED) to identify relevant trials. We will search all databases from their inception to the present. For a detailed search strategy for all electronic databases, see S2 File in Supporting information.

**Searching other resources.** The reference lists of relevant publications will be checked for any unidentified randomised trials, and authors of included studies will be contacted by email asking for unpublished randomised trials. Further, we will search for ongoing trials on the following databases:

- ClinicalTrials.gov (www.clinicaltrials.gov)

- Google Scholar (https://scholar.google.dk/)

- The Turning Research into Practice (TRIP) Database (https://www.tripdatabase.com/)

- The World Health Organization (WHO) International Clinical Trials Registry Platform (ICTRP) search portal (http://apps.who.int/trialsearch/)

- Cochrane Database of Systematic Reviews https://www.cochranelibrary.com/cdsr/about-cdsr

We will consider accessing and reading conference abstracts from conferences in child and adolescent psychiatry and psychology, autism and autism intervention (e.g., ESCAP (European Society for Child and Adolescent Psychiatry) Congresses, Autism-Europe International Congress and INSAR (International Society for Autism Research). Additionally, we will consider relevant-for-the-review unpublished and grey literature trials, if identified. Finally, we will consult content experts for relevant studies.

## Data collection and analysis

The review will follow the recommendations of the Cochrane Collaboration [90]. The analyses will be performed using Trial Sequential Analysis [91] and Stata version 17 [92].

**Selection of studies.** Two authors (CEC and SMTZ) will independently screen titles and abstracts. All relevant full-text study reports/publications will be retrieved, and two review authors (CEC and SMTZ) will independently screen the full text and identify and record reasons for excluding ineligible studies. The two authors will resolve any disagreement through discussion or (if required) by consulting a third author (MBL), and all three authors will make consensus agreements. Trial selection will be displayed in an adapted flow diagram as per the Preferred Reporting Items for Systematic Reviews and Meta-Analyses (PRISMA) statement [93].

## Data extraction and management

Two authors (CEC and SMTZ) will independently extract data from the included trials. Disagreements will be resolved by discussion with a third author (MBL) and consensus agreement made. Duplicate publications and companion trial papers will be assessed to evaluate all available data simultaneously (e.g., maximise data extraction, correct bias assessment). Software for data management will be Excel and Stata.

**Trial characteristics.** We will extract the following data: Risk of bias components (as defined below), trial design, length of follow-up (end-of-treatment or later follow-up), estimation of sample size and both inclusion and exclusion criteria.

**Participant characteristics and diagnosis.** We will extract the following data: Number of randomised participants, number of analysed participants, number of participants lost to follow-up/withdrawals/crossover, compliance with intervention, age range (mean or median), sex ratio and ethnicity.

**Parent-mediated interventions characteristics.** We will extract the following data: Type of parent-mediated intervention (behavioural, developmental, NDBI or other), treatment duration, number of sessions, session lengths (minutes) and whether treatment is given directly, online or blended.

**Comparison.** We will extract the following data about the comparison groups: Type (treatment as usual, management as usual, waitlist or no treatment).

**Outcomes.** All above-mentioned outcomes will be extracted from each randomised clinical trial, and we will identify if outcomes are incomplete or selectively reported according to the criteria described later in the "incomplete outcome data" bias domain and "selective outcome reporting" bias domain.

**Notes.** Funding of the trial and notable conflicts of interest of trial authors will be extracted if available. We will note if outcome data were not reported in a usable way in the "Characteristics of included studies" table. Two review authors (CEC and SMTZ) will independently transfer data into the Stata file [92]. Disagreements will be resolved through discussion or (if required) by consulting a third author (MBL).

## Assessment of risk of bias in included studies

The instructions given in the *Cochrane Handbook for Systematic Reviews of Interventions* [90] will be used in our evaluation of the methodology and hence the risk of bias of the included trials will be evaluated. We will evaluate the methodology with respect to the following:

•Random sequence generation

•Allocation concealment

•Blinding of participants and treatment providers

•Blinding of outcome assessment

•Incomplete outcome data

•Selective outcome reporting

•Other risk of bias

•Overall risk of bias

### Random sequence generation.

•*Low risk*: Sequence generation was achieved by using a computer random number generator or a random number table. Drawing lots, tossing a coin, shuffling cards and throwing dice will be considered adequate if performed by an independent adjudicator.

•*Unclear risk*: Method of randomisation was not specified, but the trial we evaluated was still presented as being randomised.

•*High risk*: Allocation sequence was not randomized or only quasi-randomised. These trials will be excluded.

### Allocation concealment.

•*Low risk*: Allocation of patients was performed by a central independent unit, on-site locked computer, identical-looking numbered sealed envelopes, or containers prepared by an independent investigator.

•*Unclear risk*: Trial was classified as randomised, but the allocation concealment process was not described.

•*High risk*: Allocation sequence was familiar to the investigators who assigned participants.

**Blinding of participants and treatment providers.**

•*Low risk*: Participants and treatment providers were blinded to intervention allocation and this blinding was described.

•*Unclear risk*: Procedure of blinding was insufficiently described.

•*High risk*: Blinding of participants and treatment providers was not performed.

**Blinding of outcome assessment.**

•*Low risk of bias*: Mentioning that the outcome assessors were blinded, and this blinding was sufficiently described.

•*Unclear risk of bias*: Not mentioning if the outcome assessors in the trial were blinded or the extent of blinding was insufficiently described.

•*High risk of bias*: No blinding or incomplete blinding of outcome assessors was performed.

**Incomplete outcome data.**

•*Low risk of bias*: Missing data were unlikely to cause treatment effects to deviate from plausible values. This could be either (1) there were no dropouts or withdrawals for all outcomes or (2) the numbers and reasons for the withdrawals and drop-outs for all outcomes were clearly stated and could be described as similar to both groups. Generally, the trial is judged as at low risk of bias due to incomplete outcome data if dropouts are less than 5%. However, the 5% cut-off is not definitive.

•*Unclear risk of bias*: There was insufficient information to assess whether missing data were likely to induce bias on the results.

•*High risk of bias*: The results were likely to be biased due to missing data either because the pattern of dropouts could be described as different in the two intervention groups or the trial used improper methods in dealing with the missing data (e.g., last observation carried forward).

**Selective outcome reporting.**

•*Low risk of bias*: Protocol was published before or at the start of the trial and outcomes specified in the protocol were reported.

•*Unclear risk of bias*: No protocol was published.

•*High risk of bias*: The outcomes in the protocol were not reported.

**Other risk of bias.**

•*Low risk of bias*: Trial appears to be free of other components (for example, academic or for-profit bias) that could put it at risk of bias.

•*Unclear risk of bias*: Trial may or may not be free of other components that could put it at risk of bias.

•*High risk of bias*: Other factors that could put the trial at risk of bias (for example, authors conducted trials on the same topic, for-profit bias).

**Overall risk of bias.**

•*Low risk of bias*: Trial will be classified as overall "low risk of bias" only if all the bias domains described in the above paragraphs are classified as low risk of bias.

•*High risk of bias*: Trial will be classified as "high risk of bias" if any of the bias risk domains described above are classified as "unclear" or high risk of bias.

We will assess the domains blinding of outcome assessment, incomplete outcome data, and selective outcome reporting for each outcome result. We can thus assess the bias risk for each outcome assessed in addition to each trial. Our primary conclusions will be based on the results of our primary outcome with an overall low risk of bias. Our primary and secondary conclusions will be presented in the summary of findings tables.

### Differences between protocol and the review

We will conduct the review according to the present review's published protocol and report any deviations from it in the "Differences between the protocol and the review" section of the systematic review.

### Measures of treatment effect

The mean differences (MDs) with 95% confidence interval (CI) will be calculated for the continuous primary and secondary outcomes. When there is more than one outcome measure in the exploratory outcomes, the standardised mean difference (SMD) with 95% CI for continuous outcomes will be calculated. Trial sequential analysis-adjusted confidence intervals will also be calculated (see below).

### Dealing with missing data

As the first option, we will contact all trial authors to obtain any relevant missing data (i.e., for data extraction and for the assessment of risk of bias, as specified above). We will use intention to treat data if these data are available.

### Continuous outcomes

We will primarily analyse scores assessed at single time points. If only changes from baseline scores are reported, we will analyse the results together with follow-up scores [90]. If standard deviations (SDs) are not reported, the SDs using trial data will be calculated if possible. We will use intention-to-treat data if provided by the trialist. We will not consider last observation carried forward as intention to treat data. We will not impute missing values for any outcomes in our primary analysis. We will impute data in our sensitivity analysis (see paragraph below) for continuous outcomes.

### Assessment of heterogeneity

We will primarily investigate forest plots to visually assess any sign of heterogeneity. We will secondly assess the presence of statistical heterogeneity by the chi$^2$ test (threshold $P<0.10$) and measure the quantities of heterogeneity by the $I^2$ statistic [90,94,95]

### Assessment or reporting biases

If ten or more trials are included, we will use a funnel plot to assess reporting bias. Funnel plots will be visually inspected to assess risk of bias. We are aware of the limitations of a funnel plot (i.e., a funnel plot assesses bias due to small sample size). From this information, we will assess possible reporting bias. For continuous outcomes the regression asymmetry test [96] and adjusted rank correlation [97] will be used.

### Unit of analysis issues

We will only include randomised clinical trials. For trials using crossover design, only data from the first period will be included [90,98], preventing any unit of analysis issues. We will not include cluster randomised trials.

### Data synthesis

The meta-analysis will be performed according to the recommendations stated in the *Cochrane Handbook for Systematic Reviews of Interventions* [90], Keus et al. [99], and the eight-step assessment suggested by Jakobsen et al. [100]. The

statistical software Stata version 17 [92] will be used to analyse data. We will assess the intervention effects with both random-effects meta-analyses [101] and fixed-effects meta-analyses [102]. We will consider the most conservative result (highest P-value) as the primary analysis results and the least conservative result as a sensitivity analysis [100]. A total of seven primary and secondary outcomes will be assessed, and thus a P-value of 0.013 or less will be considered statistically significant [100]. We will use the eight-step procedure to assess if the thresholds for significance are crossed [100]. Our primary conclusion will be based on the study results with low risk of bias [100]. Where multiple trial arms are reported in a single trial, we will include the relevant arms only. If two comparisons are combined in the same meta-analysis, we will divide the control group to avoid double-counting [90]. If quantitative synthesis is not appropriate due to considerable heterogeneity or a small number of included trials, we will report the results narratively.

## Trial sequential analysis

Traditional meta-analyses run the risk of random errors due to sparse data and repetitive testing of accumulated data when updating reviews. We will perform Trial Sequential Analysis to control for the risks of type I errors and type II errors. The Trial Sequential Analysis will be performed on the outcomes to calculate the required information size (i.e., the number of participants needed in a meta-analysis to detect or reject a certain intervention effect) and the cumulative $Z$-curve's breach of relevant trial sequential monitoring boundaries [103–110]. A more detailed description of trial sequential analysis can be found in the trial sequential analysis manual [103] and at http://www.ctu.dk/tsa/.

For continuous outcomes, the observed SD, a mean difference of the observed SD/2, an alpha of 1.3% for all outcomes, a beta of 20% and the observed diversity as suggested by the trials in the meta-analysis will be used in the Trial Sequential Analysis.

Estimation of the minimally clinically important differences (MCID) for the primary and secondary outcomes:

- For the ADOS total score [60] MCID was estimated 3.0 based on a previous study [71].

- Regarding Vineland Adaptive Behavior Scales a calculation [62] of Vineland-II Adaptive Behavior composite score MCID estimates range from 2.01 to 3.2 for distribution-based methods [111]. We estimate the MCID to be 3.0 for this review.

- For the Negative Effects Questionnaire [61], MCID was not calculated as it has not, to the authors knowledge, been applied to this population previously.

- For Mullen Scales of Early Learning [64], MCID was estimated to be 5.0 based on a previous intervention study for toddlers with ASD [71].

- For the parent-rated Pediatric Quality of Life Inventory [63], MCID was estimated to be 7.4 based on a previous study of quality of life in children with ASD [112].

- For the World Health Organization Quality of Life Assessment [65], MCID was estimated to be 0.3 based on a previous study in parents of children with ASD [113].

- For the parent-rated questionnaire Parenting Stress Index [66], MCID was estimated to be 10.4 based on a previous intervention study for children with ASD [114].

## Subgroup analysis

We will perform the following subgroup analyses when analysing the primary outcome of autism symptom severity:

1. High risk of bias trials compared to low risk of bias trials

2. Types of parent-mediated interventions: behavioural, developmental and NDBI interventions

3. The control groups: usual care, waiting list or no treatment

4. Single parents versus two parents

5. Younger (aged less than 7 years) versus older (with an age range including 7 years and older) children

6. Severity of ASD

We will use the formal test for subgroup interactions in Stata [92].

**"Summary of findings" table**

A summary of findings table will be created using the prespecified primary outcome, i.e., autism symptom severity as well as secondary outcomes, i.e., child adaptive functioning, adverse effects, child language, child's quality of life (assessed by parents), parental quality of life and parental stress. Finally, the exploratory outcomes will be presented in a second summary of findings table, i.e., disruptive behaviour, joint attention, child adaptive functioning, parent synchrony/sensitivity, attachment, parent fidelity to implementation of the intervention, family life functioning, autism symptom severity, child socio-communicative symptoms and child repetitive symptoms. We will use the five GRADE considerations (bias risk of the trials, consistency of effect, imprecision, indirectness, and publication bias) to assess the quality of a body of evidence as it relates to the studies which contribute data to the meta-analyses for the pre-specified outcomes [100,115–117]. We will assess imprecision using Trial Sequential Analysis. Additionally, we will use methods and recommendations described in the *Cochrane Handbook for Systematic Reviews of Interventions* [90] using GRADE pro software. We will justify all decisions to downgrade the quality of studies using footnotes and make comments to aid the reader's understanding of the review where necessary. Firstly, we will present our results in the Summary of Findings table based on the results from the trials with low risk of bias, and secondly, the results based on all trials.

**Ethics and dissemination**

Ethical approval was not required for this protocol and systematic review. Data collection for the review and meta-analysis is ongoing. The results of the systematic review are expected to be published in a peer-reviewed journal later this year to help inform future research and clinical practice.

## Discussion

This protocol aims to synthesise the current evidence of the effects, both positive and negative, of PMI on younger children with ASD and their parents. The primary outcome will be autism symptom severity. Secondary outcomes will be child adaptive functioning, serious adverse effects, child language, child socio-communicative symptoms, child's quality of life (assessed by parents) and parental quality of life. Finally, the exploratory outcomes will be disruptive behaviour, joint attention, child adaptive functioning, parent synchrony/sensitivity, attachment, parent fidelity to implementation of intervention and family life functioning.

The protocol has several strengths. Firstly, the protocol is reported in accordance with the PRISMA-P statement [58,59]. The high quality pre-defined methodology is based on the *Cochrane Handbook for Systematic Reviews of Interventions* [90], the eight-step assessment suggested by Jakobsen et al. [100], Trial Sequential Analysis [91], PRISMA guidelines [118] and GRADE assessment [115–117]. A further strength of this protocol is that it provides an overview of the effect of parent-mediated interventions for younger children with autism spectrum disorder and their parents (and families) with a pragmatic comparison to treatment as usual, waiting list or no treatment. There is an expectation to include a large number of trials, with the possibility of showing a substantial range of the effects of the parent-mediated interventions. There will be a direct emphasis on autism symptomatology.

The review also has some limitations. The primary limitation is a potential risk of large heterogeneity when including trials with all types of PMI. PMI derived from different theoretical paradigms and with different intervention goals will be included if they are parent-mediated. It is very likely that different interventions have different effects. Hence, if a significant effect of the interventions appears in the meta-analysis, it will be challenging to conclude what caused the effect. Therefore, the inclusion of sub-group analyses is important to demonstrate the effect of different kinds of intervention types. However, we may ultimately decide that a meta-analysis is not warranted. Another limitation is the inclusion of a large number of outcomes and the expected variety of assessment methods of outcomes in the different trials. In addition, the large number of comparisons increases the risk of type 1 errors. The threshold for significance is adjusted according to the number of outcomes. This substantial risk of type 1 errors will be addressed when discussing the results of the review. Also, the protocol states an intention to compare low risk of bias trial results to high risk of bias trial results, however, based on evidence from previous reviews [40,46], the field of parent-mediated interventions for children with ASD is mainly characterised by high risk of bias trials. We hope the review will disclose more recent trials with low risk of bias methodology. If the size of low risk of bias trials compared to high risk of bias are too different to make a meaningful sub-group analysis, this will not be done. Finally, the use of this pragmatic methodology will lead to the inclusion of the most relevant trials.

## Supporting information

**S1 File.  PRISMA-P Checklist.**
(PDF)

**S2 File.  Search Strategy.** Search strategies for 'Parent-mediated interventions versus usual care in children with autism spectrum disorders'.
(PDF)

## Acknowledgments

The authors wish to acknowledge librarian Sarah Klingenberg of the Copenhagen Trial Unit and librarian Jette Frost Jepsen of the Medical Library, Aalborg University Hospital who contributed to the search strategy.

## Author contributions

**Conceptualization:** Charlotte Engberg Conrad, Sonja Martha Teresa Ziegler, Niels Bilenberg, Jens Christiansen, Birgitte Fagerlund, Pia Jeppesen, Caroline Barkholt Kamp, Per Hove Thomsen, Janus Christian Jakobsen, Marlene Briciet Lauritsen.

**Funding acquisition:** Charlotte Engberg Conrad, Sonja Martha Teresa Ziegler, Niels Bilenberg, Pia Jeppesen, Marlene Briciet Lauritsen.

**Methodology:** Charlotte Engberg Conrad, Sonja Martha Teresa Ziegler, Jens Christiansen, Rikke Hermann Jakobsen, Pia Jeppesen, Caroline Barkholt Kamp, Janus Christian Jakobsen, Marlene Briciet Lauritsen.

**Project administration:** Charlotte Engberg Conrad, Janus Christian Jakobsen.

**Resources:** Janus Christian Jakobsen, Marlene Briciet Lauritsen.

**Supervision:** Niels Bilenberg, Per Hove Thomsen, Janus Christian Jakobsen, Marlene Briciet Lauritsen.

**Validation:** Charlotte Engberg Conrad, Caroline Barkholt Kamp, Janus Christian Jakobsen, Marlene Briciet Lauritsen.

**Writing – original draft:** Charlotte Engberg Conrad, Sonja Martha Teresa Ziegler.

**Writing – review & editing:** Charlotte Engberg Conrad, Sonja Martha Teresa Ziegler, Niels Bilenberg, Jens Christiansen, Birgitte Fagerlund, Rikke Hermann Jakobsen, Pia Jeppesen, Caroline Barkholt Kamp, Per Hove Thomsen, Janus Christian Jakobsen, Marlene Briciet Lauritsen.

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
