## [Decision Letter · Decision Letter 0]

26 Feb 2025

PONE-D-25-02938Parent-mediated interventions versus usual care in children with autism spectrum disorders. A protocol for a systematic review with meta-analysis and Trial Sequential Analysis.PLOS ONE

Dear Dr. Conrad,

Thank you for submitting your manuscript to PLOS ONE. After careful consideration, we feel that it has merit but does not fully meet PLOS ONE’s publication criteria as it currently stands. Therefore, we invite you to submit a revised version of the manuscript that addresses the points raised during the review process.

We look forward to receiving your revised manuscript.

Kind regards,

Simone Varrasi

Academic Editor

PLOS ONE

Journal Requirements:

2. Thank you for stating the following financial disclosure: [This work was supported by The Lundbeck Foundation, grant number R389-2021-1597, TrygFonden, grant number 155050, Novo Nordisk Foundation, grant number NNF22SA0081180, Karen Elise Jensen Foundation, and the Simon Fougner Hartmann Family Fund. The foundations did not influence development of the protocol.].

Reviewers' comments:

Reviewer's Responses to Questions

**Comments to the Author**

1. Does the manuscript provide a valid rationale for the proposed study, with clearly identified and justified research questions?

Reviewer #1: Yes

Reviewer #2: Yes

2. Is the protocol technically sound and planned in a manner that will lead to a meaningful outcome and allow testing the stated hypotheses?

Reviewer #1: Yes

Reviewer #2: Yes

3. Is the methodology feasible and described in sufficient detail to allow the work to be replicable?

Reviewer #1: Yes

Reviewer #2: Yes

4. Have the authors described where all data underlying the findings will be made available when the study is complete?

Reviewer #1: Yes

Reviewer #2: No

5. Is the manuscript presented in an intelligible fashion and written in standard English?

Reviewer #1: Yes

Reviewer #2: Yes

6. Review Comments to the Author

You may also provide optional suggestions and comments to authors that they might find helpful in planning their study.

Reviewer #1: This is a well constructed protocol with a clear justification for the trial, a review of literature, the methods and discussion. The strengths and weaknesses are also clearly described

Reviewer #2: Dear authors,

your aim to do this systematic review is valuable. To improve clarity and comprehensiveness, the following revisions are recommended

1. The definition of usual care varies widely—does it include behavioral therapy, speech therapy, or other structured interventions? Consider adding a sentence explaining how heterogeneous usual care conditions will be handled.

2. Clarifying the specific limitations of previous reviews could strengthen the justification for the study.

3. The manuscript does not specify whether grey literature or unpublished studies will be included.

4. Specify criteria for age-based subgroups. What age cut-offs define “younger” and “older” children?

7. PLOS authors have the option to publish the peer review history of their article (what does this mean? ). If published, this will include your full peer review and any attached files.

**Do you want your identity to be public for this peer review?** For information about this choice, including consent withdrawal, please see our Privacy Policy .

Reviewer #1: No

Reviewer #2: No

---

## [Author Response · Author response to Decision Letter 1]

13 Mar 2025

Comments for the academic editor:

1. I have edited the manuscript including file naming following the requirements from PLOS ONE.

2. I have added the following to the Coverletter:” This work was supported by The Lundbeck Foundation, grant number R389-2021-1597, TrygFonden, grant number 155050, Novo Nordisk Foundation, grant number NNF22SA0081180, Karen Elise Jensen Foundation, and the Simon Fougner Hartmann Family Fund. The funders had no role in study design, data collection and analysis, decision to publish, or preparation of the manuscript.”

3. It was misleading to state that I would make data available on acceptance, when I completed the data availability statement of the submission form. As this is a protocol for a future review there is no data. For the future review all data will be publicly available. The text should have been “No datasets were generated or analysed during the current study. All relevant data will be made available upon study completion”. I apologise for this inconvenience.

4. I have reviewed the Reference list and have found that it is complete and correct.

Comments for reviewer #1

Thank you so much for taking the time to review our protocol for a review with meta-analysis.

Comments for reviewer #2

Thank you so much for taking the time to review our protocol for a review with meta-analysis.

1. Thank you for pointing this out. “Usual care” is indeed a concept that has challenged us. It is widely used within the trials, as it seems unethical to limit children from receiving potential beneficial help, which they would have had, if they did not participate in the trial.

I have added the following to the Methods l. 274-276: “In usual care control groups parents are usually not restricted from having their child participate in other not parent-mediated interventions of their choice e.g., speech therapy or behavioural therapy.”

2. Thank you for suggesting further clarification of limitations for previous reviews. I decided to report on some of the items from the AMSTAR tool to demonstrate some of the limitations.

I have added the following to l. 207-213: “The reviews all had performed study selection in duplicate (AMSTAR item 5, Table 2) and used appropriate measures for statistical combination of results (item 11, Table 2). However, several of the other AMSTAR items were not fulfilled by some of the reviews, e.g., registration of a prespecified protocol (item 2, Table 2), providing a list of excluded studies and justifying exclusion (item 7, Table 2), reporting on sources for funding in included studies (item 10, Table 2), providing explanation for any heterogeneity observed in the results (item 14, Table 2).”

3. In our manuscript l. 342-350 we have the following statement regarding unpublished and grey literature: “We will consider accessing and reading conference abstracts from conferences in child and adolescent psychiatry and psychology, autism and autism intervention (e.g., ESCAP (European Society for Child and Adolescent Psychiatry) Congresses, Autism-Europe International Congress and INSAR (International Society for Autism Research). Additionally, we will consider relevant-for-the-review unpublished and grey literature trials, if identified. Finally, we will consult content experts for relevant studies.”

4. Thank you so much for pointing this out. We have now specified the age cut-offs for younger and older children in l. 544-545 “Younger (aged less than 7 years) versus older (with an age range including 7 years and older) children.”

---

## [Decision Letter · Decision Letter 1]

15 Apr 2025

Parent-mediated interventions versus usual care in children with autism spectrum disorders. A protocol for a systematic review with meta-analysis and Trial Sequential Analysis.

PONE-D-25-02938R1

Dear Dr. Conrad,

We’re pleased to inform you that your manuscript has been judged scientifically suitable for publication and will be formally accepted for publication once it meets all outstanding technical requirements.

Kind regards,

Simone Varrasi

Academic Editor

PLOS ONE

Additional Editor Comments (optional):

Reviewers' comments:

Reviewer's Responses to Questions

**Comments to the Author**

1. Does the manuscript provide a valid rationale for the proposed study, with clearly identified and justified research questions?

Reviewer #2: Yes

2. Is the protocol technically sound and planned in a manner that will lead to a meaningful outcome and allow testing the stated hypotheses?

Reviewer #2: Yes

3. Is the methodology feasible and described in sufficient detail to allow the work to be replicable?

Reviewer #2: Yes

4. Have the authors described where all data underlying the findings will be made available when the study is complete?

Reviewer #2: Yes

5. Is the manuscript presented in an intelligible fashion and written in standard English?

Reviewer #2: Yes

6. Review Comments to the Author

You may also provide optional suggestions and comments to authors that they might find helpful in planning their study.

Reviewer #2: Dear authors,

thank you for the amendments done, now the paper is ready to be accepted in the current form

7. PLOS authors have the option to publish the peer review history of their article (what does this mean? ). If published, this will include your full peer review and any attached files.

**Do you want your identity to be public for this peer review?** For information about this choice, including consent withdrawal, please see our Privacy Policy .

Reviewer #2: No

---

## [Editor Report · Acceptance letter]

PONE-D-25-02938R1

PLOS ONE

Dear Dr. Conrad,

I'm pleased to inform you that your manuscript has been deemed suitable for publication in PLOS ONE. Congratulations! Your manuscript is now being handed over to our production team.

Kind regards,

on behalf of

Dr. Simone Varrasi

Academic Editor

PLOS ONE